# Relapses Children’s Acute Lymphoblastic Leukemia, Single Center Experience

**DOI:** 10.3390/children9121874

**Published:** 2022-11-30

**Authors:** Weronika Stolpa, Magdalena Zapała, Bartosz Zwiernik, Agnieszka Mizia-Malarz

**Affiliations:** 1Department of Oncology, Hematology and Chemotherapy, Upper Silesia Children’s Care Heatlh Centre, 16 Medykow Street, 40-752 Katowice, Poland; 2Student Scientific Club, Medical University of Silesia, 40-055 Katowice, Poland; 3Department of Pediatrics, Upper Silesia Children’s Care Heatlh Centre, Medical University of Silesia, 16 Medykow Street, 40-752 Katowice, Poland

**Keywords:** acute lymphoblastic leukemia, relapse, children

## Abstract

The prognosis in children and adolescents with relapsed ALL, despite intensive therapy, including hematopoietic stem cell transplantation, is still challenging. This study aims to analyze the incidence of relapsed ALL and survival rates in correlation to the risk factors. Materials and methods: 125 pediatric patients with ALL diagnosed in our department between 2000-2018; age 1–18 years old (median 6.4); female 53.6% vs. male 46.4%. Results: 19 pts (15.2%) were diagnosed with a relapse. Three pts (15.8%) had been diagnosed with very early relapses (2/3 T-ALL), 12 pts (63.1%) as an early relapse, and 4 pts (21.1%) as a late relapse. Bone marrow was the most frequent relapses localization. The five-year survival has been achieved by six patients (31.6%). A significant difference was found in regard to the five-year overall survival and relapse type (*p* < 0.05). The group with very early relapses (3/3; 100%) has not reached the five-year survival. Conclusions: 1. The main prognostic factor in children’s ALL relapses is still the time of the onset of the relapse. 2. The T lineage acute lymphoblastic leukemia is a worse prognostic factor. 3. The analysis of the above relapse risk factors alongside cytogenethic markers and flow cytometry or polymerase chain reaction minimal residual disease is very important for first-line chemotherapy improvement and a more personalized choice of therapy for ALL patients.

## 1. Introduction

Acute lymphoblastic leukemia (ALL) is the most common malignant disease in the pediatric population [1,2,3]. The outcomes of treating children and adolescents with ALL in Poland are satisfying. According to the Polish Pediatric Leukemia/Lymphoma Study Group (PPLLSG), the five-year event-free survival is approximately 83%, which is a result of the applied treatment-group risk-adjusted multidrug chemotherapy protocols [1,4]. However, 15–20% of children with ALL have relapsed, despite the high-intensive chemotherapy [5,6,7,8,9]. The prognosis in children and adolescents with relapses of ALL, despite intensive therapy, including hematopoietic stem cell transplantation (HSCT), is still challenging—only about 30–50% of these children can be cured [8].

This study aims to analyze the incidence of relapsed ALL and survival rates in correlation to the risk factors in patients treated in our center.

## 2. Materials and Methods

This is a retrospective observational study of pediatric patients with newly diagnosed ALL and admitted to the Oncology, Hematology, and Chemotherapy Department of the Upper Silesia Children’s Healthcare Center between 2000 and 2018. We analyzed 125 patients aged between 1 to 18 years old (median age of 6.4). The majority of patients were female (53.6% vs. 46.4%). ALL diagnosis was based on the results of bone marrow aspiration biopsy (infiltration ≥ 25% blast cells in bone marrow smears). ALL subtypes were confirmed through immunophenotypic analysis. Moreover, prognostically significant cytogenetic abnormalities were searched in every patient included in the group. All of the patients received first-line treatment according to an ALL-BFM 95, ALL IC 2002, and ALL IC BFM 2009 chemotherapy protocols (depending on the year of the diagnosis and obligatory treatment protocol in Poland). Children were classified into risk groups: standard, intermediate and high according to features such as age at diagnosis, immunophenotyping, cytogenetic examination results, white blood cells (WBC) count at diagnosis, response to the seven-day steroid therapy, and results of the bone marrow evaluation at the 15 and 33 days of induction treatment. The maintenance treatment was based on applied chemotherapy protocols [1,5].

The clinical characteristics, information about chemotherapy treatment, and transplantation procedures were collected from the medical records of our center. The definition of relapse after the first complete remission was infiltration above 5% of blast cells in the bone marrow and/or blasts in extra-medullary sites. Relapse classification defined as: “very early” (VER), “early” (ER), and “late” (LR) was based on the time of relapse after initial diagnosis-under 18 months, between 18 and 36 months, and above 36 months, respectively. The cut-off point of follow-up was defined as August 2021.

### Statistical Analysis

STATISTICA 13.0 was used for data analysis. Student’s *t*-test and ANOVA tests were used for comparing numerical data. The chi-squared tests were used for comparing non-numerical dates. *p*-values of <0.05 were considered significant. The overall survival (OS) was measured based on the time of diagnosis until the patients’ death or their last documented follow-up visit. The progress-free survival (PFS) was based on the end of maintenance therapy until patients’ relapse (relapse diagnosis date). The Kaplan–Meier estimates were used to determine overall survival (OS).

## 3. Results

One hundred twenty-five newly diagnosed ALL patients (group A) were retrospectively enrolled in this study. Moreover, group A was divided into group B-patients who achieved complete remission (CR) after the first line of treatment (*n* = 106; 84.8%) and continued to do so, and group C-patients who have been diagnosed with relapses (*n* = 19; 15.2%). The above three groups were analyzed. In all groups, the median age was similar (between 6.4 and 6.6). The gender distribution (M/F) was comparable in groups A (1.16/1) and B (1/1); in the group C were significantly more girls (0.38/1). The most common diagnosis in all groups was B-line ALL, a common subtype. In each group, the highest number of patients belonged to the intermediate-risk group. Detailed characteristics that compare groups (A, B, and C) are presented in Table 1.

Our analysis of group C showed that twelve of the relapsed patients (63.1%) had “early” relapse, which was the most common type, followed by “late” (4 patients, 21.1%) and “very early” (3 patients, 15.8%). The results are shown in Table 2. Bone Morrow was the most frequent site of relapse (52.6%), followed by the isolated central nervous system (CNS) (36.8%). In one case, relapse was noticed in the testicles. Moreover, one patient manifested complex localization of relapse: involvement of testicular and bone morrow. Additionally, B-line ALL was more frequent than T-line ALL (subtype pre-T), which was confirmed only in three children (15.7%).

Several congenital genetic abnormalities have been linked to the relapsed group. Down Syndrome (constitutive chromosome 21 trisomy) was detected in one case. Moreover, one patient was observed with numeric chromosomal loss (hypodiploidy), and two patients represented a group of genetic abnormalities (12, 21) with ETV6-RUNX1 fusion.

Considering the mean periods from obtaining CR1 to the onset of relapse, we notice that T-ALL has a shorter period than B-ALL (15.7 months vs. 34.3 months). If we start our analysis from the localization of relapsed, we could see that the mean period for bone morrow relapsed was 34 months, and, for central nervous system localization, 23 months. What is more, among patients with T-ALL, mean overall survival (OS) amounted to 23 months versus 60 months for B-ALL. Comparing the OS in terms of recurrence localization-a slightly better OS was obtained in the group of extramedullary (55.6 months) versus myeloid (52.7 months) localization.

Relapsed ALL patients were treated according to the current treatment protocol. Protocol ALL REZ BFM 2002 was applied in 9 patients, and IntReALL 2010 protocol in 10 patients. Moreover, eight children underwent allogeneic hematopoietic stem cell transplantation (allo-HSCT), and one patient with LR passed the allo-HSCT procedure, preceded by autologous hematopoietic stem cell transplantation (auto-HSCT). In one case (myeloid, ER ALL), the chimeric antigen receptor T cell therapy (CAR T-cell) procedure was successfully applied.

From the relapsed ALL group, five-year survival has been achieved by six patients (31.6%). The VER group (3/3; 100%) has not reached the five-year survival. If we analyzed five-year OS in ER ALL, we could see that in 75% (9/12) cases, five-year survival has not been achieved; on the other hand, in the LR ALL group, only 25% (1/4) children have not reached five-year survival. A significant difference was found in regard to the five-year OS and relapse type (*p* < 0.05). The overall survival was found to be 97.0 months for LR ALL, 47.2 months for ER patients, and the worst OS for VER was 25.7 months. The graph (Figure 1) shows the correlations between the incidence of disease relapse and patients’ survival after the recurrence of ALL. The event-free time for all relapsed patients was in the range of up to maximum 81 months. Twelve patients experienced a relapse before completing maintenance treatment. Unfortunately, 13 (68.4%) relapsed ALL patients died. A large majority—76.9% (10/13)—of these deaths were due to disease progression. The reason for the rest, three deaths was a complication after the allo-HSCT procedure. The overall survival median—from ALL diagnoses to death—was 44 months (range: 10–147), and from relapse, ALL diagnoses totaled 19 months (range 0–59 months).

## 4. Discussion

Despite the vast improvement in the outcome after up-front, ALL treatment, the relapse, and associated treatment failure are still challenging. In most available analyses, the relapse rate oscillated between 15 and 20% [4,5,6]. In our clinic number of relapses achieved 15.2%, which is compatible with the above-mentioned range. According to reports, patients with relapse were stratified as high-risk at initial diagnosis [10,11,12]. What is more, Tuong et al. showed that relapsing in HRG patients was more frequent by 1,6 times than in IRG (61.5 vs. 38.5%) [13]. However, the result of our analysis does not confirm this knowledge—in our study group, a relapse was most often diagnosed in patients who qualified basically for intermediate risk. This is probably the result of a relatively small study group or/and a fairly short period of patient observation.

Moreover, the T-cell immunophenotype is one of the main individual prognostic factors [10]. In this study, out of 10 children with T-ALL, 3 patients experienced a relapse. According to Nguyen et al., most often, localization of an isolated relapse is bone marrow (50–60%), next CNS (20%), and testicular (5%) [10]. These results are close to our study. The most common localization in our patients was bone marrow (52.6%), least often in testicular (5.3%).

By assessing the correlation between experience and the type of relapse, we showed that the overall survival (OS) median for LR was the highest (97 months). Moreover, five-year survival indicates an improved response rate in patients with “late” relapse. All VER patients have not achieved five years of survival, which can be indicated as the worst prognostic.

Our results support new studies, which named the time from diagnosis to relapse and localization the strongest risk factor for overall survival. Previously, Gaynon et al. reported that the strongest predictor of prolonged survival after relapse was an initial time to first relapse ≥ 36 months. What is more, patients with an isolated bone marrow relapse who failed initial therapy less than 18 months from diagnosis had a higher risk for death [12]. The latest dates clearly define localization and time of relapse as the most serious risk factors for overall survival. Moreover, Nguyen et al. determined time of relapse was the greatest factor in five-year survival. In the mentioned date, five-year survival had a high value with LR patients (VER 11.49% vs. ER 18.42% vs. LR 43.46%) [10].

The cytogenetic risk factors are significant markers [1,2,3,4,9,10]. With an increased understanding of the ALL, cytogenetics is expected to be included in risk stratification. From patients with relapse, we noticed one child with Down Syndrome and one patient with hypodiploidy. Moreover, in 2 cases, t(12,21) was shown, which is considered a favorable prognostic factor. The presented retrospective analysis concerns the patients with diagnosed lymphoblastic leukemias in our Center in the past (2000–2018). The small number of demonstrated cytogenetic abnormalities in leukemia cells may result from the imperfections of the methods used in diagnostics at that time.

Oskarsson et al., in the large, retrospective analysis of children with ALL from 1992–2011 in the Scandinavian population, suggest indicating the existence of individual prognostic factors of relapse. These are unfavorable cytogenetics, Down syndrome and additionally age under 10 years, and T-cell lineage immunophenotype with hyperleukocytosis [7].

Huang et al. suggest in their model the existence of new biomarkers for the early recurrence of B-cell ALL in pediatric patients. The expression of the genes HOXA7, S100A10, and S100A11 and low expression of IFI44L is associated with poor prognosis and chemoresistance in high-risk patients. Moreover, specific MiRNAs as novel markers of relapse in ALL are still investigating [11].

Recent studies on the genomic background of relapsed ALL propose that the leukemia evolution is not a single-line process but more branched with new, secondary genetic alternations. This phenomenon may lead to a reduction in the chemotherapy sensitivity of leukemia cells [9]. Despite improvements in the first-line therapy of pediatric ALL, identification of the high-risk relapse group is still unsatisfactory, and because of that, searching for new biomarkers is necessitated.

At the time of the follow-up (08.2021) from relapsed ALL group, 13 patients (68.4%) died, while 6 patients (31.6%) are still alive. The median time between relapse and death was 19 months (range: 0–59 months) and was higher than the other results [10,13]. In our opinion, this may be due to a smaller group of patients included in our retrospective analysis. The causes of death in our group, disease progression, and complications after allogeneic HSCT confirm the fact that leukemia cells are resistant to relapses. Therefore, more therapeutic options are being searched. The promising treatment option for relapsed B-cell lineage ALL is CAR T-cell therapy. Tisgenlecleucel (chimeric antigen receptor anti-CD19 with T lymphocytes) indicates high response rates among patients with relapsed B-cell ALL [14]. The evidence of the effectiveness of CAR T-cell therapy is one patient from our center with a second relapse of pre-B ALL and chemoresistance-obtained complete remission, and for more than a year, this patient has been in remission. Unfortunately, relapse of T-cell lineage ALL is still challenging-the effective and safe therapy should be researched.

This paper presented a single-center retrospective study of ALL cases in children focusing on the analysis of known risk factors for ALL relapse. Newly introduced prognostic factors such as flow cytometry minimal residual disease (FC MRD), polymerase chain reaction minimal residual disease (PCR MRD), and blast cells cytogenetic diagnostics alongside commonly analyzed factors enable an improvement in monitoring of residual disease and results of the first-line chemotherapy in leukemia as well as a more personalized and comprehensive choice of therapy for ALL patients.

## 5. Conclusions

The main prognostic factor in children’s ALL relapses is still the time of the onset of the relapse.The T lineage acute lymphoblastic leukemia is worse prognostic factor.There is a strong need for analysis of the relapse risk factors alongside cytogenetic markers and prognostic factors such as flow cytometry minimal residual disease or polymerase chain reaction minimal residual disease (PCR) for improvement of first-line chemotherapy in leukemia and more personalized choice of therapy for ALL patients.

## Figures and Tables

**Figure 1 children-09-01874-f001:**
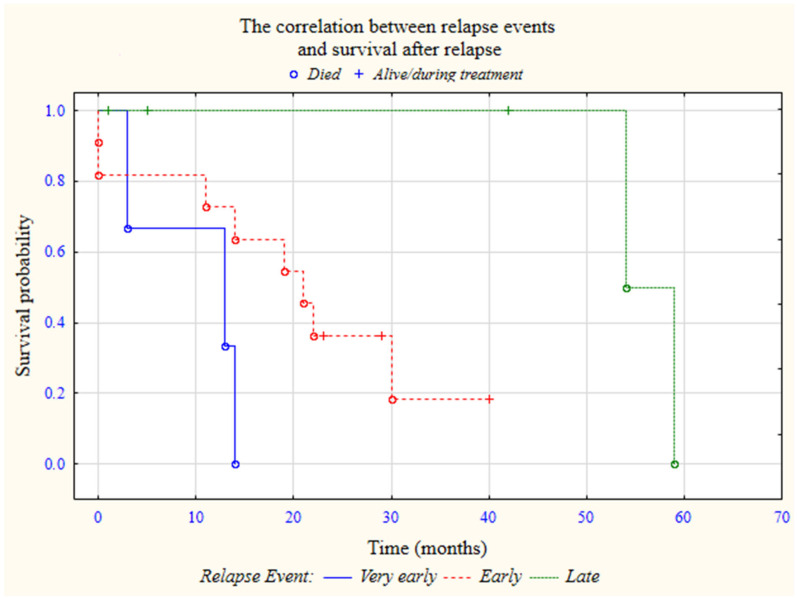
The correlation between relapse events and survival after relapse.

**Table 1 children-09-01874-t001:** The clinical characteristic comparing the three selected groups with acute lymphoblastic leukemia.

Kerrypnx		Group A Studied Patients (*n* = 125)	Group B Non-Relapsed ALL Patients (*n* = 106)	Group C Relapsed ALL Patients (*n* = 19)
Age at diagnosis (years)	mean ± SD median (IQR) <10 years ≥10 years	6.4 ± 4.3 5.0 (1.5–18.00) 103 (82.4%) 22 (17.6%)	6.3 ± 4.3 5.0 (1.5–18.00) 87 (82.1%) 19 (17.9%)	6.6 ± 4.4 5.0 (2.0–17.0) 16 (84.2%) 3 (15.8%)
Sex	Male Female	67 (53.60%) 58 (46.40%)	53 (50%) 53 (50%)	5 (26.32%) 14 (73.68%)
Type of leukemia	B-cell T-cell	115 (92.0%) 10 (8.0%)	99 (93.40%) 7 (6.60%)	16 (84.21%) 3 (15.79%)
Subtype of leukemia	Common Pre-B Pro-B Pre-T	76 (60.8%) 38 (30.4%) 1 (0.80%) 10 (8.0%)	67 (63.2%) 32 (30.2%) 0 (0%) 7 (6.6%)	9 (47.4%) 6 (31.6%) 1 (5.3%) 3 (15.7%)
Risk group	SRG IRG HRG	40 (32.0%) 56 (44.8%) 29 (23.2%)	36 (34.0%) 45 (42.5%) 25 (23.6%)	4 (21.1%) 11 (57.9%) 4 (21.1%)
Genetics	Ph(+) DS	11 (8.8%) 5(4.0%)	11 (10.4%) 4 (3.8%)	0 (0.0%) 1 (5.3%)

SRG—standard-risk group. IRG—intermediate-risk group. HRG—high-risk group. DS—Down syndrome.

**Table 2 children-09-01874-t002:** Characteristics of group C.

	Group C (*n* = 19, 100%)			
	VER ALL (*n* = 3; 15.5%)	ER ALL (*n* = 12; 63.2%)	LR ALL (*n* = 4; 33.3%)	*p*
BFM classification	S4: 2 (66.7%) S2: 1 (33.3%)	S3: 6 (50.0%) S2: 6 (50.0%)	S1: 1 (25.0%) S2: 3 (75.0%)	-
B cell ALL T cell ALL	1 (33.3%) 2 (66.7)	11 (91.7%) 1 (8.3%)	4 (100.0%) 0 (0.0%)	<0.05
Localization	BM 2 (66.7%) T 1 (33.3%)	BM 5 (41.7%) T 1 (8.3%) CNS 5 (41.7%) BM + T 1 (8.3%)	BM 3 (75.0%) CNS 1 (25.0%)	>0.05
Allo HSCT	2 (66.7%)	5 (41.7%)	1 (25.0%)	-
Allo HSCT + Auto HSCT			1 (25.0%)	
OS (mth) Mean ± SD Median (IQR)	25.7 (10.0–39.0) 28 (±14.6)	47.2 (17.0–77.0) 45.5 (±17.0)	97 (38.0–147.0) 101.5 (±44.8)	<0.05
PFS (mth)	0	3.6 (0–35.0)	29.8 (7.0–81.0)	<0.05
Five years OS	0 (0%)	3 (25.0%)	3 (75.0%)	>0.05
Death	3 (100%)	8 (66.7%)	2 (50.0%)	>0.05

Legend: VER—very early relapsed ALL. ER—early relapsed ALL. LR—late relapsed ALL. BFM—Berlin-Frankfurt-Munster Group, Germany. HSCT—Host stem cell transplantation. OS—Overall survival. PFS—Progression-free survival. BM—bone marrow. T-testis. CSN—central nervous system.

## Data Availability

The data supporting reported results can be found in our hospital archive.

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
