# Peer review of "Relapses Children’s Acute Lymphoblastic Leukemia, Single Center Experience"

_children, 2022, doi:10.3390/children9121874_

Round 1

Reviewer 1 Report

Please replace commas with decimal point when dealing with numbers  (for instance 76.9% instead of 76,9%).

Lines 83-84 - Please change to - "In each group, the highest number of patients 'belonged' to the intermediate-risk group."

Line 94 - Is it twenty or twelve (number of patients with early relapse). Table 2 shows n=12 for early relapse.

 Lines 151-152 - It would be prudent to add references to support this statement. 

Author Response

REV answers 1

Dear Reviewer 1,

Thank you so much for reading the manuscript called:

Relapses children’s acute lymphoblastic leukemia, single centre experience

Weronika Stolpa, Magdalena Zapała, Bartosz Zwiernik, Agnieszka Mizia-Malarz

Your comments and remarks  are of great value to me. The corrections made according to your tips will improve the substantive quality of the manuscript and make it of higher transparency.

Ad1.

Please replace commas with decimal point when dealing with numbers  (for instance 76.9% instead of 76,9%).

I changed it in the manuscript.

Ad.2

Lines 83-84 - Please change to - "In each group, the highest number of patients 'belonged' to the intermediate-risk group."

I changed it in the manuscript.

Ad.3

Line 94 - Is it twenty or twelve (number of patients with early relapse). Table 2 shows n=12 for early relapse.

Twelve (12) of course.

Ad.4

Lines 151-152 - It would be prudent to add references to support this statement. 

We added references.

I would like to emphasize that the necessity of corrections in accordance with your suggestions enriched my work experience. I appeal to you with the hope of getting a positive opinion  and acceptance of my manuscript. Your opinion is of great importance to me.

Sincerely,

Agnieszka Mizia-Malarz

Reviewer 2 Report

1.       This paper should be rejected on the basis of no IRB reporting. The authors are reporting patient data, and therefore an IRB approval or waver must be obtained and reported to ethically report or publish these data.

2.       Informed consent is applicable as patient data is being reported. Also, the Data Availability Statement is also applicable and has not been completed.

3.       The general applicability of these data is not clear. Most data agree with what has been published, in line 157 the authors state their results “do not confirm knowledge” or, more specifically, do not support that higher risk groups are more prone to relapse than intermediate risk groups. This should be expanded.

4.       In line 180, the authors state that cytogenetic risk factors are “really important”. Two points, first, the writing should be more professional and, second, the only cytogenetics mentioned in the retrospective analysis are broad chromosomal abnormalities (Ph status, trisomy 21, hypodiploidy) and risk groups that are only defined as standard, intermediate, and high with no attention to the actual cytogenetics used to define these groups. More information regarding cytogenetics on the reported patient population would be necessary to make this argument.

5.       The numbers in the data tables are not aligned (eg Table 1, subtype, table 2 auto, auto +allo, table 2 mean and median OS). This makes the numbers difficult to understand.

Author Response

REV answers 2

Dear Reviewer 2,

Thank you so much for reading the manuscript called:

Relapses children’s acute lymphoblastic leukemia, single centre experience

Weronika Stolpa, Magdalena Zapała, Bartosz Zwiernik, Agnieszka Mizia-Malarz

Your comments and remarks  are of great value to me. The corrections made according to your tips will improve the substantive quality of the manuscript and make it of higher transparency.

Ad1.

This paper should be rejected on the basis of no IRB reporting. The authors are reporting patient data, and therefore an IRB approval or waver must be obtained and reported to ethically report or publish these data.

Our manuscript has been supplemented with this statement:

The consent of the bioethics committee was revoked (it was not needed) due to retrospective nature of the work ,which is in line with the regulations of our Institutional Review Board. This work  was created on the basis of our patients’ test results  obtained during routine examinations in the course of the treatment.

Ad.2

Informed consent is applicable as patient data is being reported. Also, the Data Availability Statement is also applicable and has not been completed.

This work is based on the results of the patients’  examinations that we obtained during their

treatment in the years 2000-2018. According to our Institutional regulations Informed Consent

Station in retrospective studies/ examinations is not needed.

Data Availability Statement:

The results of the research used in the work are stored in our hospital archives

Ad.3

The general applicability of these data is not clear. Most data agree with what has been published, in line 157 the authors state their results “do not confirm knowledge” or, more specifically, do not support that higher risk groups are more prone to relapse than intermediate risk groups. This should be expanded.

I added:

This may be due to a relatively small research group and / or a fairly short period of time

observation of the  patients included in the study in recent years.

Ad.4

In line 180, the authors state that cytogenetic risk factors are “really important”. Two points, first, the writing should be more professional and, second, the only cytogenetics mentioned in the retrospective analysis are broad chromosomal abnormalities (Ph status, trisomy 21, hypodiploidy) and risk groups that are only defined as standard, intermediate, and high with no attention to the actual cytogenetics used to define these groups. More information regarding cytogenetics on the reported patient population would be necessary to make this argument.

The presented retrospective analysis concerns the patients with diagnosed  llymphoblastic leukemias in our Center in the past (2000-2018). The small number of demonstrated cytogenetic abnormalities in leukemia cells may result from the imperfections of the methods used in diagnostics at that time.

This paragraph has been corrected.

Ad 5.

The numbers in the data tables are not aligned (eg Table 1, subtype, table 2 auto, auto +allo, table 2 mean and median OS). This makes the numbers difficult to understand.

The data tables have been aligned.

I would like to emphasize that the necessity of corrections in accordance with your suggestions enriched my work experience. I appeal to you with the hope of getting a positive opinion  and acceptance of my manuscript. Your opinion is of great importance to me.

Sincerely,

Agnieszka Mizia-Malarz

Round 2

Reviewer 1 Report

I commend you for making necessary changes as suggested.

Author Response

REV answers 1

Dear Reviewer 1,

Thank you so much for reading the manuscript called:

Relapses children’s acute lymphoblastic leukemia, single centre experience

Weronika Stolpa, Magdalena Zapała, Bartosz Zwiernik, Agnieszka Mizia-Malarz

Thank you for more positive opinion after corrections. Now I made some minor adjustments in the results.

I would like to emphasize that the necessity of corrections in accordance with your suggestions second time enriched my work experience. I appeal to you with the hope of getting a positive opinion  and acceptance of my manuscript. Your opinion is of great importance to me.

Sincerely,

Agnieszka Mizia-Malarz

Reviewer 2 Report

The quality of the manuscript has been greatly improved. Minor typos and awkward sentence structure still exists, but the content is appropriate for publication.

Author Response

REV answers 2

Dear Reviewer 2,

Thank you so much for reading the manuscript called:

Relapses children’s acute lymphoblastic leukemia, single centre experience

Weronika Stolpa, Magdalena Zapała, Bartosz Zwiernik, Agnieszka Mizia-Malarz

Thank you for more positive opinion after corrections. Now I made some minor adjustments in the results and conclusion.

I would like to emphasize that the necessity of corrections in accordance with your suggestions enriched second time my work experience. I appeal to you with the hope of getting a positive opinion  and acceptance of my manuscript. Your opinion is of great importance to me.

Sincerely,

Agnieszka Mizia-Malarz